# Clinical Outcomes and Predictors of Mortality in Patients with Difficult-to-Treat Resistant *Pseudomonas aeruginosa* Infections: A Retrospective Cohort Study

**DOI:** 10.3390/antibiotics15010033

**Published:** 2026-01-01

**Authors:** Alberto Enrico Maraolo, Antonella Gallicchio, Vincenzo Fotticchia, Maria Rosaria Catania, Riccardo Scotto, Ivan Gentile

**Affiliations:** 1Section of Infectious Diseases, Department of Clinical Medicine and Surgery, University of Naples “Federico II”, 80138 Naples, Italy; gallicchioantonella9@gmail.com (A.G.); vincenzo.fotticchia@unina.it (V.F.); ri.scotto@gmail.com (R.S.); ivan.gentile@unina.it (I.G.); 2VIII Infectious Disease Unit, P.O. Cotugno, A.O.R.N. dei Colli, 80131 Naples, Italy; 3Department of Molecular Medicine and Medical Biotechnologies, University of Naples Federico II, 80131 Napoli, Italy; mariarosaria.catania@unina.it

**Keywords:** difficult-to-treat resistant *Pseudomonas aeruginosa*, 30-day mortality, septic shock, ceftazidime-avibactam, ceftolozane-tazobactam, source control, clinical outcomes

## Abstract

**Background:** Difficult-to-treat resistant *Pseudomonas aeruginosa* (DTR-PA) infections are associated with high morbidity and mortality, but data on prognostic factors remain limited. Given the limited real-world data on outcomes of DTR-PA infections, we aimed to identify clinical predictors of mortality and response to therapy in this setting. **Methods:** We conducted a single-center retrospective cohort study of 51 patients with DTR-PA infections. The primary endpoint was 30-day all-cause mortality; secondary endpoints were clinical and microbiological cure at end of therapy. An exploratory analysis evaluated 30-day infection-related mortality. Logistic regression models (univariable, multivariable and Firth bias-reduced) were used to identify independent predictors. **Results:** Median age was 64 years (IQR 22); 63% were male and 71% were in the ICU at infection onset. Sepsis occurred in 80% and septic shock in 45%. Thirty-day all-cause mortality was 49% (25/51). According to multivariable analysis, septic shock was an independent predictor of mortality (aOR 5.52, 95% CI 1.04–29.27; *p* = 0.045) as younger age (aOR 1.06, 95% CI 1.00–1.12; *p* = 0.052), whereas targeted therapy with ceftazidime/avibactam or ceftolozane/tazobactam is a protective factor (aOR 0.15, 95% CI 0.02–1.17; *p* = 0.070) did not reach significance in the final model. Clinical cure occurred in 33% (17/51) and was negatively associated with device burden and bloodstream infection, whereas microbiological cure (45%, 23/51) was more likely with targeted therapy and absence of sepsis. The exploratory analysis of infection-related mortality (35%) showed similar predictors. **Conclusions:** DTR-PA infections are associated with high mortality. Septic shock and older age predict death, while the use of novel β-lactam/β-lactamase inhibitors is associated with improved outcomes. Early recognition of severe illness and timely administration of active therapy may improve survival in these infections.

## 1. Introduction

*Pseudomonas aeruginosa* is a major cause of healthcare-associated infections (HAIs), particularly in critically ill and immunocompromised patients [1]. Due to its significant impact on public health, *P. aeruginosa* is recognized as a critical priority pathogen by global health organizations [2].

The emergence of strains non-susceptible to all first-line agents, including carbapenems—termed difficult-to-treat resistant *P. aeruginosa* (DTR-PA)—represents a critical therapeutic challenge [3,4,5]. These infections are associated with limited treatment options, high healthcare costs and substantial morbidity and mortality [6,7].

The global prevalence of DTR-PA infections among clinical isolates is estimated at 2–17%, with significant regional variation and higher rates in certain high-risk settings [8]. These rates are generally lower than those for multidrug-resistant (MDR) or carbapenem-resistant *P. aeruginosa* (CRPA), which can be higher than 50% in some regions [9].

Despite the increasing clinical use of newer antipseudomonal agents such as ceftazidime/avibactam (C/A) and ceftolozane/tazobactam (C/T), data regarding outcomes of DTR-PA infections and their predictors remain scarce [10,11]. In this study, we investigated a cohort of patients with DTR-PA infections, to evaluate predictors of 30-day all-cause mortality (primary endpoint) as well as clinical and microbiological cure at the end of therapy (secondary endpoints). Additionally, we explored 30-day infection-related mortality.

## 2. Results

51 patients with DTR-PA infections were included in the analysis. The overall median age was 64 years (interquartile range [IQR] 53–75) and 32 (63%) were male. The majority of patients (71%) were hospitalized in the ICU at the onset of infection. The median Charlson Comorbidity Index (CCI) was 8.0 (IQR 5.0). At clinical presentation, 80% of patients had sepsis and 45% had septic shock. The most common type of infection was ventilator-associated pneumonia (VAP) (65%). As detailed in Table 1, patients who did not survive to 30 days were significantly older than survivors (median age 72 vs. 60 years, *p* = 0.018). Mechanical ventilation was more frequent among non-survivors (88% vs. 50%, *p* = 0.003), and the same applied for sepsis (100% vs. 62%, *p* < 0.001) and septic shock (76% vs. 15%, *p* < 0.001). Furthermore, the median Sequential Organ Failure Assessment (SOFA) score was significantly higher in the non-survivor group (11.0 vs. 4.0, *p* < 0.001).

### 2.1. Primary Outcome: 30-Day All-Cause Mortality

The primary outcome of 30-day all-cause mortality occurred in 25 of 51 patients (49%).

In the univariable analysis, factors significantly associated with higher mortality included septic shock (odds ratio (OR) 17.42; 95% confidence interval [CI] 4.27–71.07; *p* < 0.001), mechanical ventilation (OR 7.33; 95% CI 1.75–30.66; *p* = 0.006) and VAP (OR 10.77; 95% CI 1.18–98.03; *p* = 0.035). An increase in the SOFA score was also associated with increased odds of death (OR 1.45; 95% CI 1.19–1.77; *p* < 0.001). Other variables with a *p* < 0.20 were carried forward to the multivariable analysis.

After model selection based on the Aikake Information Criterion (AIC), the final multivariable model retained four key predictors (Table 2). Using Firth’s bias-reduced regression, the independent predictors for 30-day mortality were:Septic Shock: (adjusted OR [aOR] 5.52; 95% CI 1.04–29.27; *p* = 0.045);Age: (aOR 1.06; 95% CI 1.00–1.12; *p* = 0.052);Targeted Therapy, namely Receiving “Other” targeted therapy (vs. no targeted therapy) was associated with lower mortality (aOR 0.13; 95% CI 0.02–1.02; *p* = 0.052);Delta (Δ)SOFA Score: (aOR 1.19; 95% CI 0.96–1.47; *p* = 0.113).

The final model demonstrated excellent discrimination, with an Area Under the Curve (AUC) of 0.918 (95% CI 0.846–0.990) (Figure 1 and Appendix A).

### 2.2. Secondary Outcomes

#### 2.2.1. Clinical Cure at End of Therapy

Clinical cure was achieved in 17 of 51 patients (33.3%). In the final Firth-corrected multivariable model, factors independently associated with a higher likelihood of clinical cure were (Table 3):Absence of Catheter-Related Bloodstream Infection (CRBSI): patients with CRBSI had significantly lower odds of achieving clinical cure (aOR 0.04; 95% CI 0.00–0.60; *p* = 0.020).Lower number of invasive devices: an increasing number of devices was associated with lower odds of cure (aOR 0.38; 95% CI 0.15–0.93; *p* = 0.035).

The model for clinical cure showed outstanding discrimination with an AUC of 0.933 (95% CI 0.857–1.000) [See Appendix A].

#### 2.2.2. Microbiological Cure at End of Therapy

Microbiological cure was observed in 23 of 51 patients (45.1%). The final Firth-corrected multivariable model identified two main predictors (Table 4):Absence of Sepsis: patients with sepsis had significantly lower odds of microbiological cure (aOR 0.19; 95% CI 0.04–0.90; *p* = 0.036).Targeted therapy: receiving C/A or C/T was associated with higher odds of achieving microbiological cure compared to no targeted therapy (aOR 4.63; 95% CI 0.96–22.18; *p* = 0.055).

The model’s discrimination was good, with an AUC of 0.803 (95% CI 0.677–0.929) [See Appendix A].

In Figure 2 an OR plot is depicted to describe in a more comprehensive manner the variables impacting on the three above-mentioned outcomes and retained in the final model featuring Firth correction.

### 2.3. Exploratory Outcome (30-Day Infection-Related Mortality)

Death was attributed to the DTR-PA infection in 18 of 51 patients (35.3%). The final multivariable model with Firth correction for this outcome identified the following predictors:Septic Shock: (aOR 7.02; 95% CI 1.57–31.36; *p* = 0.011).Targeted therapy: both treatment groups showed significantly lower odds of infection-related mortality compared to no targeted therapy.C/A or C/T: (aOR 0.03; 95% CI 0.00–0.30; *p* = 0.003).Other targeted therapies: (aOR 0.08; 95% CI 0.01–0.58; *p* = 0.013).

This model also demonstrated high discrimination with an AUC of 0.892 (95% CI 0.801–0.985).

See Appendix A for univariable and multivariable analyses as well as for model diagnostics (Appendix A, respectively).

## 3. Discussion

In this cohort of patients with DTR-PA infections, we observed a high 30-day all-cause mortality rate of 49%, confirming the severe clinical impact of this pathogen. This finding is consistent with U.S. Centers for Disease Control and Prevention (CDC) estimates regarding the lethality of *P. aeruginosa* healthcare-associated infections (HAIs) [12]. Similarly, European Centre for Disease Prevention and Control (ECDC) data indicate that *P. aeruginosa* is the most common pathogen in ICU-acquired pneumonia, with approximately one in three ICU-acquired HAIs contributing to death [13]. While our observed mortality is higher than that reported in general estimates—such as a nationwide Italian study reporting ~19% attributable mortality for CRPA bloodstream infections (BSIs), whereas the crude death rate was 32.8% [14]—it aligns closely with series focusing on high-acuity settings. For instance, multicenter analyses of CRPA infections have reported mortality rates rising from ~39% in general CRPA cohorts to ~50% specifically within DTR-PA subsets: mortality was strongly linked to organ failure severity and improved with receipt of appropriate therapy [15].

Our higher case-fatality rate likely reflects the confluence of difficult-to-treat resistance profiles and a critical-illness case mix, characterized by a high prevalence of ICU admission and septic shock.

Host severity played a central role in prognosis. In our multivariable models, septic shock emerged as the strongest independent predictor of mortality (aOR ≈ 5.52 for all-cause death, aOR ≈ 7.02 for infection-related death), along with increasing age. While the baseline SOFA score was associated with mortality in univariable analysis, the ΔSOFA did not retain significance in the final model. Crucially, the use of active targeted therapy was associated with survival. Although the association between ‘Other’ active targeted therapies and all-cause mortality fell just short of statistical significance (aOR 0.13, *p* = 0.052), the signal was robust in the infection-related mortality model. In that analysis, receipt of targeted therapy (whether C/A, C/T, or other active agents) was associated with significantly lower odds of death compared to receiving no targeted therapy: aOR 0.03, (*p* = 0.003) for C/A or C/T and aOR 0.08, (*p* = 0.013) for ‘Other’ targeted therapies. However, given the limited sample size and wide confidence intervals, these results should be interpreted as suggestive of a beneficial effect rather than definitive proof of efficacy.

A particularly relevant clinical question is the choice between C/T and C/A [10]. Our sample size precluded a robust multivariable comparison between these specific agents; however, descriptive data indicated that C/T was used as a backbone therapy significantly more frequently in survivors than in non-survivors (38% vs. 8%, *p* = 0.029). While previous studies, such as the multicenter analysis by Hareza et al., found no significant mortality difference between C/T and C/A [16], more recent data from the large multicenter CACTUS study reported higher clinical success rates with C/T, particularly in pneumonia, though mortality rates remained comparable [17]. Similarly, a multicenter cohort study by Almangour et al. corroborated these findings, showing comparable efficacy and safety profiles for both agents, while highlighting that C/T may offer advantages in specific clinical scenarios [18]. These observations may be supported by pharmacokinetic/pharmacodynamic (PK/PD) data: C/T achieves robust ELF penetration with favorable time > Minimum Inhibitory Concentration (MIC) at approved 3 g q8h dosing and AUC of 50% and 62% for ceftolozane and tazobactam, respectively [19], whereas PK studies in healthy volunteers receiving C/A ceftazidime–avibactam 2 g/0.5 g or 3 g/1 g every 8 h showed that ELF-to-plasma concentration ratios were approximately 23–26% for ceftazidime and 28–35% for avibactam, indicating ELF exposures of roughly one third of those in plasma [20]. While our data prevents drawing firm conclusions, these findings collectively suggest that C/T may offer advantages in specific high-burden respiratory infections.

Regarding secondary endpoints, microbiological cure was achieved in 45% and absence of sepsis plus receipt of targeted therapy (with C/A or C/T) were important predictors, though statistical significance was borderline (*p* ≈ 0.055). For infection-related mortality (35.3%), septic shock again was a key risk factor, whereas targeted therapy (both C/A or C/T and other active agents) was associated with significantly lower odds of death.

Our analysis of secondary endpoints highlighted the critical role of device burden. Indeed, clinical cure at end of therapy was achieved in ~33% of patients. Patients with multiple invasive devices or CRBSI had substantially lower odds of achieving clinical cure. This underscores that antimicrobial therapy alone is often insufficient; effective source control is essential. This is particularly relevant given that routine clinical practice often lacks standardized biofilm susceptibility testing, making physical removal of colonized devices a priority as per current guidelines [21,22,23].

In the context of monotherapy versus combination therapy, evidence remains heterogeneous across carbapenem-resistant Gram-negative infections [24] and specifically for VAP by *P. aeruginosa*, where studies report conflicting results: whereas some suggested benefit with combinations [25,26], others showed no significant advantage over monotherapy [22,27]. Notably, our cohort consisted predominantly of critically ill patients (80% with sepsis, 45% with septic shock and 65% with VAP) and targeted monotherapy was administered only in a minority of patients (35%). In the light of very heterogeneous regimens as for monotherapy and combinations, we did not test this variable in the regression models. Notably, mortality was higher in subjects undergoing combination therapies (61%—20 out of 33—vs. 28%—5 out of 18). While this unadjusted finding might seem to favor monotherapy, it is likely confounded by indication, as patients in septic shock may have been more likely to receive combination regimens. Current literature suggests that susceptibility-confirmed monotherapy (e.g., with a novel anti-pseudomonal agent) is likely adequate even in critically ill patients, provided there is timely initiation and PK/PD optimization. Decisions should remain individualized, considering patient severity, site of infection and local resistance epidemiology, and combinations reserved for selected scenarios (e.g., very high inoculum while awaiting susceptibilities, delayed source control or metallo-β-lactamase producers requiring C/A plus aztreonam). Larger, prospective studies are needed to confirm these signals in high-acuity DTR-PA populations.

Clinical implications from our findings and the literature suggest that in DTR-PA infections: early and appropriate targeted therapy is crucial; C/T may provide an advantage over C/A in selected infection sites (particularly pneumonia) or patient subgroups, though the risk of emergent resistance should be monitored; monotherapy may be adequate also in patients with sepsis [21,22]; device burden is a critical and modifiable barrier to cure, emphasizing the need for timely source control.

This study has nevertheless several limitations. First, its retrospective and single-center design limits the generalizability of findings and carries inherent risks of selection bias. Second, the cohort included only 51 patients. This small sample size resulted in a low events-per-variable ratio for the regression analyses. Although we applied Firth’s penalized likelihood correction to mitigate bias and handle separation (infinite estimates), the models remain underpowered, and the confidence intervals for some estimates are wide. Consequently, the reported associations should be interpreted with caution. Third, our cohort was predominantly critically ill (high ICU prevalence with frequent sepsis, septic shock and ventilator-associated pneumonia), which likely inflated baseline mortality risk and may limit external validity to mixed-acuity populations despite statistical adjustment. Fourth, heterogeneity in therapeutic strategies over the study period may have influenced results: novel β-lactam/β-lactamase inhibitors such as C/T and C/A were not consistently available or routinely tested in the early years of the study, potentially introducing treatment bias and limiting a more granular microbiological assessment via the assessment of MICs distributions. Fifth, device burden was evaluated as number of invasive devices but without standardized information on duration of device exposure or biofilm assessment, which may underestimate its impact. Sixth, emergence of resistance could not be adequately studied, as repeat cultures were limited and cefiderocol testing was not performed; resistance mechanisms or standardized assessments for biofilm formation on invasive devices were not performed were not addressed as well in detail. Finally, unmeasured confounders, such as detailed immune status, PK/PD optimization and adequacy of supportive care, may also have influenced outcomes.

## 4. Materials and Methods

### 4.1. Study Design and Population

We conducted a retrospective cohort study including consecutive adult patients diagnosed with invasive DTR-PA infections at our institution. The study population comprised patients admitted between January 2018 and December 2023, with infection onset either in the ICU or other hospital wards. The reporting of the study followed the Strengthening the Reporting of Observational Studies in Epidemiology (STROBE) Statement guidelines [28].

Eligible cases were identified through a review of clinical microbiology records. Inclusion criteria required: (1) isolation of *P. aeruginosa* from blood, urine, or respiratory cultures; (2) a resistance profile meeting the DTR criteria (defined as mentioned in the proper paragraph); and (3) a confirmed clinical diagnosis of infection based on concurrent clinical, radiological, and laboratory findings. Across the study period, 568 adult *P. aeruginosa* infection episodes were screened; among these, 51 patients met the strict DTR inclusion criteria and constituted the final analytic cohort.

### 4.2. Data Collection

Clinical and demographic data, underlying comorbidities, infection types, invasive devices and therapeutic management were retrospectively collected from patients’ electronic medical records using a standardized extraction form.

Collected variables included:Demographics & Comorbidities: Age, sex, ward of hospitalization at onset, and underlying comorbidities (summarized using the Charlson Comorbidity Index [CCI]).Healthcare Exposures: Prior hospitalization, surgery, antimicrobial therapy, immunosuppressive therapy, or documented *P. aeruginosa* infection within the preceding 3 months.Immunocompromised Status: Defined as active chemotherapy for malignancy, solid organ or hematopoietic stem cell transplant, absolute neutrophil count < 500 cells/µL, or receipt of immunosuppressive therapy within 90 days prior to infection.Clinical Presentation: Sepsis or septic shock status at onset, SOFA score, and the presence of invasive devices (central venous catheter, urinary catheter, mechanical ventilation, chest drainage tube).Infection Characteristics: Site of infection (ventilator-associated pneumonia, hospital-acquired pneumonia, bloodstream infection, catheter-related bloodstream infection, complicated urinary tract infection) and polymicrobial status.Therapeutic Management: Infectious diseases consultation, source control procedures, empirical therapy (regimens, adequacy, duration), targeted therapy (regimens, timing), and combination therapy use.

### 4.3. Microbiological Testing

Antimicrobial susceptibility testing was performed according to the European Committee on Antimicrobial Susceptibility Testing (EUCAST) criteria in effect at the time of isolation [29]. Routine susceptibility testing included ciprofloxacin, piperacillin–tazobactam, ceftazidime, cefepime, meropenem, amikacin, gentamicin and colistin. C/T and C/A were not routinely tested during the early years of the study and were assessed only in selected cases. Cefiderocol was not tested at any time during the study period.

### 4.4. Definitions and Outcomes

Difficult-to-treat resistant *Pseudomonas aeruginosa* (DTR-PA) was defined according to the Infectious Diseases Society of America (IDSA) criteria as *P. aeruginosa* isolates non-susceptible to all standard first-line antipseudomonal β-lactams (piperacillin–tazobactam, ceftazidime, cefepime, aztreonam, meropenem, imipenem–cilastatin and doripenem) and fluoroquinolones [22]. Sepsis was defined as life-threatening organ dysfunction resulting from a dysregulated host response to infection, identified by an increase in the SOFA score ≥ 2 points from baseline [30]. Septic shock was defined as sepsis with persisting hypotension requiring vasopressors to maintain a mean arterial pressure ≥ 65 mmHg and a serum lactate concentration > 2 mmol/L despite adequate fluid resuscitation [30]. The variable Δ (Delta) SOFA was defined as each increment of one point beyond baseline value.

Appropriate empiric therapy was defined as administration of at least one antipseudomonal agent to which the isolate was subsequently shown to be susceptible. Targeted therapy referred to treatment guided by antimicrobial susceptibility testing and administered with guideline-concordant dosing and duration. Combination therapy was defined as the concurrent use of two or more antipseudomonal agents. Time to effective therapy was defined as the interval between collection of the index culture and initiation of an active antimicrobial agent. Polymicrobial infection was defined as the isolation of at least one additional non-contaminant pathogen from the same infection episode, while recurrence was defined as the re-isolation of *P. aeruginosa* with an identical susceptibility profile within 30 or 90 days after documented clearance of the primary infection. Adequate source control was defined as device removal or surgical intervention at the infection site, including drainage of any collections, when applicable.

The assessed outcomes are described and defined below.

Primary outcome: 30-day all-cause mortality after infection onset.Secondary outcomes: (i) clinical cure at end of therapy and (ii) microbiological cure at end of therapy.Exploratory outcome: 30-day infection-related mortality.

Thirty-day mortality was defined as all-cause death occurring within 30 days of the index culture. Clinical cure was defined as the complete resolution of clinical signs and symptoms present at infection onset without the need to change or prolong antimicrobial therapy due to treatment failure or toxicity, whereas clinical failure was defined as persistence or progression of infection manifestations despite appropriate therapy. Microbiological cure was defined as the documented eradication (no growth) of the initial pathogen from follow-up cultures and microbiological failure as the persistence of the same pathogen in repeat cultures from the same site. Infection-related mortality was defined as death occurring in the presence of a persistently positive DTR-PA culture and/or unequivocal clinical or laboratory signs of infection at the time of death.

### 4.5. Statistical Analysis

Descriptive statistics were used to summarize the study population. Continuous variables were presented as medians with interquartile ranges (IQRs) and categorical variables as absolute counts and percentages. Comparisons between survivors and non-survivors were performed using Student’s t-test or the Mann–Whitney U test for continuous variables, as appropriate, and the χ^2^ test or Fisher’s exact test for categorical variables.

Following the descriptive analyses, a three-step modelling strategy was implemented. First, univariable logistic regression models were fitted for each prespecified covariate; variables with a *p*-value < 0.20 (Wald test) were carried forward to the multivariable model. Second, a multivariable logistic regression model including all candidate variables was constructed, and the final model was selected based on the lowest Akaike Information Criterion (AIC). Collinearity was assessed using the generalized variance inflation factor (GVIF) and highly collinear variables were excluded. Third, the covariates retained in the final model were re-estimated using Firth’s bias-reduced logistic regression [31].

Indeed, the small sample size resulted in a low events-per-variable (EPV) ratio, which increases the risk of overfitting and separation (manifesting as infinite parameter estimates or extremely wide confidence intervals). To mitigate these biases, the final multivariable models were estimated using Firth’s penalized likelihood method (Firth logistic regression). This approach reduces small-sample bias and provides finite parameter estimates even in the presence of separation. Consequently, results should be interpreted as suggestive of associations rather than definitive causal links.

Results were reported as odds ratios (ORs) or adjusted odds ratios (aORs) with 95% confidence intervals (CIs). The same analytic approach was applied to the secondary endpoints. Two-sided *p* < 0.05 was considered statistically significant. Model discrimination was assessed using the area under the receiver operating characteristic (ROC) curve (AUC), with 95% CIs calculated by DeLong’s method. McFadden’s pseudo-R^2^ was reported in the Supplement.

All analyses were conducted using R (version 4.4.3; R Foundation for Statistical Computing, Vienna, Austria; URL: https://www.R-project.org/).

## 5. Conclusions

In conclusion, DTR-PA infections carry a high risk of mortality. Early administration of active targeted therapy, especially with novel β-lactam/β-lactamase inhibitors (C/T and C/A), was associated with improved survival, microbiological cure and lower infection-related mortality. Clinical cure, however, remained limited (~33%), with sepsis/septic shock and invasive device burden as major barriers. While the choice between monotherapy and combination therapy remains debated, our data suggest that monotherapy can be considered also in critical patients, although treatment should be individualized based on infection site and local resistance pattern.

These findings highlight the importance of timely initiation of active therapy, comprehensive source control and individualized treatment strategies based on severity, infection site and resistance patterns.

Future multicenter prospective studies are warranted to validate these findings and to guide optimization of treatment strategies for DTR-PA infections.

## Figures and Tables

**Figure 1 antibiotics-15-00033-f001:**
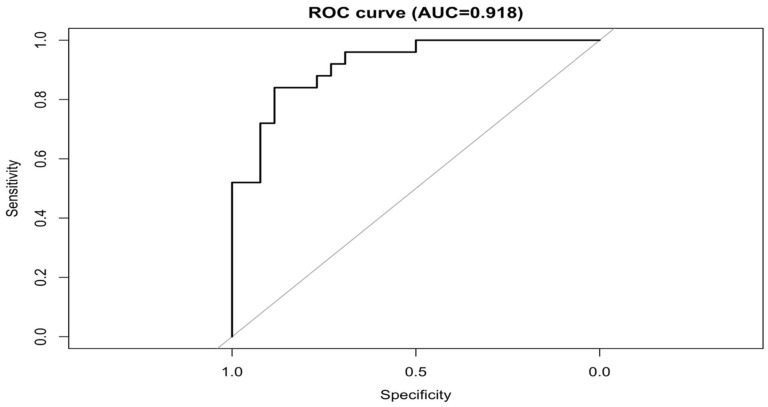
Receiver Operating Characteristic (ROC) Curve for 30-Day All-Cause Mortality Model.

**Figure 2 antibiotics-15-00033-f002:**
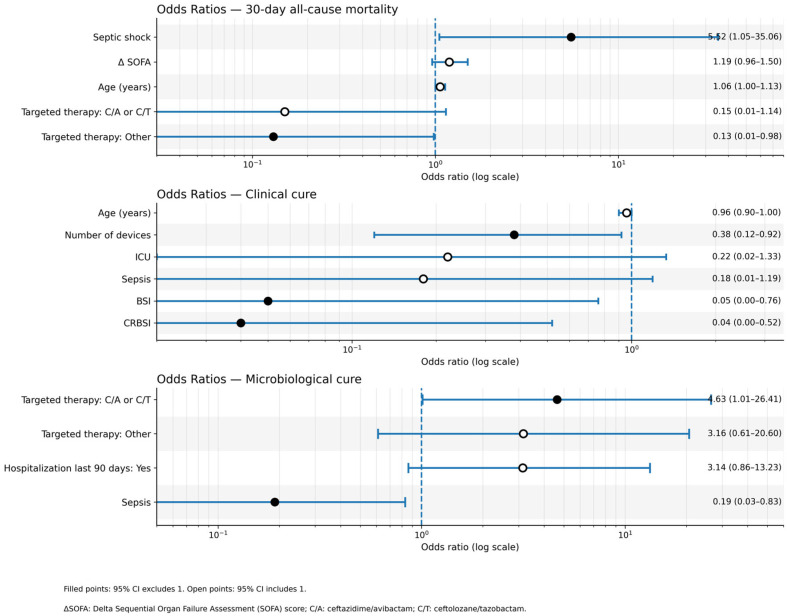
Odds ratio plot including the covariates retained in models with Firth correction assessing all-cause mortality, clinical and microbiological cure.

**Table 1 antibiotics-15-00033-t001:** Demographic and baseline characteristics of included patients according to the main outcome status.

Section	Variables	Overall (N = 51)	Alive (N = 26)	Died (N = 25)	*p*-Value
**Unit and Demographics**					
	Age, median (IQR)	64.00 (53.00–75.00)	60.00 (49.37–70.62)	72.00 (65.00–79.00)	0.018
	Male sex	32 (63%)	17 (65%)	15 (60%)	0.7
	ICU Ward of hospitalization at infection onset	36 (71%)	16 (62%)	20 (80%)	0.15
**Baseline Comorbidity and Other Relevant Factors**					
	Myocardial Infarction	11 (22%)	6 (23%)	5 (20%)	0.8
	Heart failure	4 (8%)	3 (12%)	1 (4%)	0.6
	Peripheral Vascular Disease	8 (16%)	3 (12%)	5 (20%)	0.5
	Stroke	21 (41%)	10 (38%)	11 (44%)	0.7
	Dementia	8 (16%)	4 (15%)	4 (16%)	>0.9
	Chronic Obstructive Pulmonary Disease	27 (53%)	14 (54%)	13 (52%)	0.9
	Connective Tissue Disease	4 (8%)	1 (4%)	3 (12%)	0.3
	Peptic Ulcer Disease	8 (16%)	2 (8%)	6 (24%)	0.14
	Chronic liver Disease				0.2
	Mild	14 (27%)	5 (19%)	9 (36%)	
	Moderate-to-severe	1 (2%)	0 (0%)	1 (4%)	
	Diabetes Mellitus				>0.9
	Uncomplicated	7 (14%)	4 (15%)	3 (12%)	
	End-organ damage	7 (14%)	3 (12%)	4 (16%)	
	Hemiplegia or Paraplegia	13 (25%)	8 (31%)	5 (20%)	0.4
	Chronic Kidney Disease	16 (31%)	7 (27%)	9 (36%)	0.5
	Solid Tumor				0.5
	Localized	8 (16%)	3 (12%)	5 (20%)	
	Metastatic	7 (14%)	5 (19%)	2 (8%)	
	Leukemia	4 (8%)	1 (4%)	3 (12%)	0.3
	Lymphoma	3 (6%)	1 (4%)	2 (8%)	0.6
	Charlson Comorbidity Index, median (IQR)	8.00 (5.50–10.5)	7.50 (4.87–10.12)	8.00 (6.00–10.00)	0.4
	Solid Organ Transplant	1 (2%)	0 (0%)	1 (4%)	0.5
	Severe Neutropenia (<500 cells/µL)	1 (2%)	0 (0%)	1 (4%)	0.5
**Previous Healthcare Exposure**					
	Previous hospitalization, last 3 months	32 (63%)	16 (62%)	16 (64%)	0.9
	Previous Immunosuppressive Drug use, last 3 months	13 (25%)	7 (27%)	6 (24%)	0.8
	Previous Pseudomonas Infection, last 3 months	30 (59%)	18 (69%)	12 (48%)	0.12
	Previous antibiotic therapy, last 3 months	49 (96%)	24 (92%)	25 (100%)	0.5
	Previous surgery, last 3 months	33 (65%)	16 (62%)	17 (68%)	0.6
**Type of Devices**					
	Central Venous Catheter	46 (90%)	22 (85%)	24 (96%)	0.3
	Foley Urinary Catheter	46 (90%)	22 (85%)	24 (96%)	0.3
	Mechanical Ventilation	35 (69%)	13 (50%)	22 (88%)	0.003
	Chest Drainage Tube	16 (31%)	5 (19%)	11 (44%)	0.057
**Number of Invasive Devices**					0.10
	2	5 (9.8%)	4 (15%)	1 (4%)	
	3	26 (51%)	13 (50%)	13 (52%)	
	4	12 (24%)	3 (12%)	9 (36%)	
	5	2 (4%)	1 (4%)	1 (4.0%)	
**Features on Clinical Presentation**					
	Sepsis	41 (80%)	16 (62%)	25 (100%)	<0.001
	Septic Shock	23 (45%)	4 (15%)	19 (76%)	<0.001
	Sequential Organ Failure Assessment (SOFA Score)	8.00 (3.75–12.25)	4.00 (1.00–7.00)	11.00 (7.50–14.50)	<0.001
	Sequential Organ Failure Assessment (SOFA Score), Delta	5.00 (1.00–9.00)	2.00 (0.00–4.00)	9.00 (7.50–10.50)	<0.001
	Polymicrobial Infection	44 (86%)	21 (81%)	23 (92%)	0.4
	Hospital-Acquired Pneumonia (HAP)	10 (20%)	6 (23%)	4 (16%)	0.7
	Ventilator-Associated Pneumonia (VAP)	33 (65%)	13 (50%)	20 (80%)	0.025
	Complicated Urinary Tract Infection (UTI)	3 (6%)	3 (12%)	0 (0%)	0.2
	Bloodstream Infection (BSI)	15 (29%)	7 (27%)	8 (32%)	0.7
	Catheter-Related Bloodstream Infection (CRBSI)	7 (14%)	3 (12%)	4 (16%)	0.7
**Treatment-Related Variables**					
	Infectious Diseases Consultation	46 (90%)	24 (92%)	22 (88%)	0.7
**Source Control**					0.6
	Needed but not performed	2 (3.9%)	2 (7.7%)	0 (0%)	
	Needed and performed	8 (16%)	4 (15%)	4 (16%)	
**Empirical Therapy**	Empirical therapy	41 (80%)	19 (73%)	22 (88%)	0.3
	Empirical monotherapy	25 (49%)	11 (42%)	14 (56%)	0.4
	Adequate empirical therapy	8 (16%)	4 (15%)	4 (16%)	0.5
	Duration of Empirical therapy, days (n = 41)	10.00 (5.50–14.50)	14.00 (9.50–18.50)	8.50 (5.13–11.87)	0.011
**Targeted Backbone Therapy**					0.029
	Amikacin	2 (4%)	2 (8%)	0 (0%)	
	Cefiderocol	8 (16%)	5 (19%)	3 (12%)	
	Ceftazidime/Avibactam	9 (18%)	3 (12%)	6 (24%)	
	Ceftolozane/Tazobactam	12 (24%)	10 (38%)	2 (8%)	
	Colistin	6 (12%)	2 (8%)	4 (16%)	
	Targeted monotherapy	18 (35%)	13 (50%)	5 (20%)	0.046
	Days to Active Therapy, days (n = 37)	6.00 (2.00–10.00)	8.00 (4.25–11.75)	4.00 (1.25–6.75)	0.010

ICU: intensive care unit; IQR: interquartile range.

**Table 2 antibiotics-15-00033-t002:** Univariable and Multivariable Analyses for 30-Day All-Cause Mortality.

30-Day All-Cause Mortality (Alive = 26, Died = 25)					
Variables		Univariable OR (CI 95%, *p*-Value)	Multivariable aOR (CI 95%, *p*-Value)	Final Model aOR (CI 95%, *p*-Value)	Final Model with Firth aOR (CI 95%, *p*-Value)
Age		1.05 (1.01–1.09, *p* = 0.020)	1.06 (0.99–1.14, *p* = 0.096)	1.07 (1.01–1.17, *p* = 0.049)	1.06 (1.00–1.12, *p* = 0.052)
Ward of hospitalization at infection onset	ICU	2.50 (0.71–8.80, *p* = 0.154)	0.36 (0.01–14.22, *p* = 0.589)		
Sex	Male	0.79 (0.25–2.48, *p* = 0.691)			
Charlson Comorbidity Index		1.11 (0.94–1.31, *p* = 0.223)			
Previous hospitalization, last 3 months	Yes	1.11 (0.36–3.46, *p* = 0.856)			
Previous antibiotic therapy, last 3 months	Yes	6,303,500.55 (0.00–Inf, *p* = 0.992)			
Immunodeficiency	Yes	0.92 (0.31–2.77, *p* = 0.886)			
Previous Pseudomonas Infection, last 3 months	Yes	0.41 (0.13–1.29, *p* = 0.127)	0.31 (0.02–4.28, *p* = 0.381)		
Previous surgery, last 3 months	Yes	1.33 (0.42–4.21, *p* = 0.630)			
Mechanical Ventilation	Yes	7.33 (1.75–30.66, *p* = 0.006)	7.05 (0.00–220,315.49, *p* = 0.712)		
Foley Urinary Catheter	Yes	4.36 (0.45–42.08, *p* = 0.203)			
Central Venous Catheter	Yes	4.36 (0.45–42.08, *p* = 0.203)			
Chest Drainage Tube	Yes	3.30 (0.94–11.57, *p* = 0.062)	1.77 (0.11–29.38, *p* = 0.692)		
Septic Shock	Yes	17.42 (4.27–71.07, *p* < 0.001)	8.44 (0.80–89.03, *p* = 0.076	8.04 (1.25–68.1, *p* = 0.035)	5.52 (1.04–29.27, *p* = 0.045)
Nosocomial Pneumonia (reference: no pneumonia)	HAP	4.67 (0.40–53.95, *p* = 0.217)	5.23 (0.09–319.12, *p* = 0.430)		
	VAP	10.77 (1.18–98.03, *p* = 0.035)	1.84 (0.00–70,055.10, *p* = 0.909)		
Bloodstream Infection (BSI) or Catheter-Related Bloodstream Infection (CRBSI) (reference no BSI/CRBSI)	BSI	1.90 (0.45–7.98, *p* = 0.381)			
	CRBSI	1.69 (0.33–8.73, *p* = 0.532)			
Infectious Diseases Consultation	Yes	0.61 (0.09–4.01, *p* = 0.608)			
Empirical therapy (reference: No empirical therapy)	Not adequate	2.80 (0.61–12.75, *p* = 0.183)	2.50 (0.13–47.58, *p* = 0.541)		
	Adequate	2.33 (0.34–16.18, *p* = 0.391)	1.40 (0.04–47.42, *p* = 0.853)		
Targeted therapy: Ceftazidime/Avibactam or Ceftolozane/Tazobactam vs. Other	C/A or C/T	0.25 (0.06–1.06, *p* = 0.059)	0.10 (0.00–2.97, *p* = 0.185)	0.10 (0.01–0.96, *p* = 0.067)	0.15 (0.02–1.17, *p* = 0.070)
(reference: no targeted treatment)	Other	0.31 (0.07–1.43, *p* = 0.133)	0.10 (0.00–4.69, *p* = 0.237)	0.08 (0.00–0.77, *p* = 0.046)	0.13 (0.02–1.02, *p* = 0.052)
Sequential Organ Failure Assessment (SOFA) Score, Delta		1.45 (1.19–1.77, *p* < 0.001)	1.20 (0.89–1.64, *p* = 0.236)	1.23 (0.97–1.61, *p* = 0.10)	1.19 (0.96–1.47, *p* = 0.113)
Polymicrobial Infection	Yes	2.74 (0.48–15.65, *p* = 0.257)			

**Table 3 antibiotics-15-00033-t003:** Univariable and Multivariable Analyses for clinical cure at the end of therapy.

Clinical Cure at End of Therapy (No Clinical Cure = 34, Clinical Cure = 17)					
Variables		Univariable OR (CI 95%, *p*-Value)	Multivariable aOR (CI 95%, *p*-Value)	Final Model aOR (CI 95%, *p*-Value)	Final Model with Firth aOR (CI 95%, *p*-Value)
Age		0.96 (0.92–1.00, *p* = 0.041)	0.93 (0.86–1.01, *p* = 0.087)	0.94 (0.88–1.00, *p* = 0.053)	0.96 (0.91–1.00, *p* = 0.062)
Ward of hospitalization at infection onset	ICU	0.19 (0.05–0.70, *p* = 0.012)	0.13 (0.01–3.18, *p* = 0.212)	0.11 (0.01–0.95, *p* = 0.045)	0.22 (0.04–1.24, *p* = 0.085)
Sex	Male	0.78 (0.24–2.57, *p* = 0.682)			
Charlson Comorbidity Index		0.89 (0.74–1.07, *p* = 0.208)			
Previous hospitalization, last 3 months	Yes	2.57 (0.69–9.50, *p* = 0.158)	3.04 (0.19–49.51, *p* = 0.434)		
Immunodeficiency	Yes	1.81 (0.56–5.89, *p* = 0.324)			
Previous antibiotic therapy, last 3 months	Yes	0.00 (0.00–Inf, *p* = 0.992)			
Previous Pseudomonas Infection, last 3 months	Yes	1.00 (0.31–3.26, *p* = 1.000)			
Previous surgery, last 3 months	Yes	0.32 (0.09–1.08, *p* = 0.067)	0.22 (0.01–3.70, *p* = 0.291)		
Number of Invasive Devices		0.27 (0.11–0.63, *p* = 0.003)	1.12 (0.11–11.26, *p* = 0.921)	0.26 (0.06–0.78, *p* = 0.034)	0.38 (0.15–0.93, *p* = 0.035)
Sepsis	Yes	0.07 (0.01–0.39, *p* = 0.002)	0.09 (0.00–5.24, *p* = 0.247)	0.08 (0.00–0.80, *p* = 0.056)	0.18 (0.03–1.21, *p* = 0.077)
Nosocomial Pneumonia (reference: no pneumonia)	HAP	0.90 (0.13–6.08, *p* = 0.914)	0.00 (0.00-Inf, *p* = 0.997)		
	VAP	0.13 (0.02–0.72, *p* = 0.019)	0.00 (0.00–Inf, *p* = 0.997)		
Bloodstream Infection (BSI) or Catheter-Related Bloodstream Infection (CRBSI) (reference no BSI/CRBSI)	BSI	0.14 (0.02–1.24, *p* = 0.077)	0.00 (0.00–Inf, *p* = 0.996)	0.01 (0.00–0.45, *p* = 0.086)	0.05 (0.00–1.02, *p* = 0.052)
	CRBSI	0.21 (0.02–1.95, *p* = 0.170)	0.00 (0.00–Inf, *p* = 0.996)	0.01 (0.00, 0.22, *p* = 0.020)	0.04 (0.00–0.60, *p* = 0.020)
Infectious Diseases Consultation	Yes	0.73 (0.11–4.82, *p* = 0.740)			
Empirical therapy (reference: No empirical therapy)	No	1.17 (0.25–5.41, *p* = 0.844)			
	Yes	1.40 (0.20–10.03, *p* = 0.738)			
Targeted therapy: Ceftazidime/Avibactam or Ceftolozane/Tazobactam vs. Other	C/A or C/T	1.87 (0.44–7.96, *p* = 0.394)			
(reference: no targeted treatment)	Other	0.83 (0.16–4.21, *p* = 0.825)			
Sequential Organ Failure Assessment (SOFA) Score, Delta		0.67 (0.53–0.85, *p* = 0.001)	0.93 (0.62–1.39, *p* = 0.712)		
Polymicrobial Infection	Yes	0.31 (0.06–1.61, *p* = 0.165)	0.00 (0.00-Inf, *p* = 0.997)		

**Table 4 antibiotics-15-00033-t004:** Univariable and Multivariable Analyses for microbiological cure at the end of therapy.

Microbiological Cure at End of Therapy (No Microbiological Cure = 28, Microbiological Cure = 23)					
Variables		Univariable OR (CI 95%, *p*-Value)	Multivariable aOR (CI 95%, *p*-Value)	Final Model aOR (CI 95%, *p*-Value)	Final Model with Firth aOR (CI 95%, *p*-Value)
Age		0.99 (0.96–1.03, *p* = 0.676)			
Ward of hospitalization at infection onset	ICU	0.42 (0.12–1.45, *p* = 0.172)	10.37 (0.45–237.58, *p* = 0.143)		
Sex	Male	0.44 (0.14–1.39, *p* = 0.161)	0.19 (0.02–1.44, *p* = 0.108)		
Charlson Comorbidity Index		1.13 (0.95–1.34, *p* = 0.165)	1.08 (0.78–1.50, *p* = 0.630)		
Previous hospitalization, last 3 months	Yes	3.60 (1.04–12.40, *p* = 0.042)	3.23 (0.49–21.42, *p* = 0.225)	3.65 (0.90–17.5, *p* = 0.080)	3.14 (0.84–11.72, *p* = 0.088)
Immunodeficiency	Yes	1.73 (0.57–5.28, *p* = 0.333)			
Previous antibiotic therapy, last 3 months	Yes	0.00 (0.00–Inf, *p* = 0.992)			
Previous Pseudomonas Infection, last 3 months	Yes	1.17 (0.38–3.59, *p* = 0.788)			
Previous surgery, last 3 months	Yes	0.52 (0.16–1.66, *p* = 0.270)			
Number of Invasive Devices		0.62 (0.34–1.13, *p* = 0.121)	1.29 (0.25–6.68, *p* = 0.758)		
Sepsis	Yes	0.14 (0.03–0.77, *p* = 0.023)	0.10 (0.00–2.30, *p* = 0.148)	0.14 (0.02–0.71, *p* = 0.030)	0.19 (0.04–0.90, *p* = 0.036)
Nosocomial Pneumonia (reference: no pneumonia)	HAP	1.40 (0.20–10.03, *p* = 0.738)	3.26 (0.10–107.22, *p* = 0.508)		
	VAP	0.30 (0.06–1.49, *p* = 0.141)	0.27 (0.00–37.67, *p* = 0.604)		
Bloodstream Infection (BSI) or Catheter-Related Bloodstream Infection (CRBSI) (reference no BSI/CRBSI)	BSI	0.22 (0.04–1.20, *p* = 0.081)	0.89 (0.08–10.18, *p* = 0.923)		
	CRBSI	0.67 (0.13–3.44, *p* = 0.628)	0.09 (0.01–1.59, *p* = 0.102)		
Infectious Diseases Consultation	Yes	3.67 (0.38–35.36, *p* = 0.261)			
Empirical therapy (reference: No empirical therapy)	No	2.20 (0.48–9.99, *p* = 0.309)			
	Yes	2.33 (0.34–16.18, *p* = 0.391)			
Targeted therapy: Ceftazidime/Avibactam or Ceftolozane/Tazobactam vs. Other (reference: no targeted treatment)	C/A or C/T	5.96 (1.26–28.10, *p* = 0.024)	8.60 (0.82–89.85, *p* = 0.072)	5.70 (1.10–39.0, *p* = 0.050)	4.63 (0.96–22.18, *p* = 0.055)
	Other	2.85 (0.57–14.33, *p* = 0.203)	3.48 (0.23–51.56, *p* = 0.365)	3.81 (0.64–30.1, *p* = 0.200)	3.16 (0.59–16.97, *p* = 0.179)
Sequential Organ Failure Assessment (SOFA) Score, Delta		0.83 (0.71–0.96, *p* = 0.013)	0.89 (0.65–1.21, *p* = 0.452)		
Polymicrobial Infection	Yes	0.57 (0.11–2.86, *p* = 0.494)			

## Data Availability

Anonymized data and code for running the analysis can be provided upon reasonable request.

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
