# Peer review of "Antibiotics2026, 15(1), 33;https://doi.org/10.3390/antibiotics15010033"

_antibiotics, 2026, doi:10.3390/antibiotics15010033_

Round 1

Reviewer 1 Report

Comments and Suggestions for Authors

This manuscript investigates predictors of mortality and treatment outcomes in patients with difficult-to-treat resistant Pseudomonas aeruginosa infections. The topic is clinically important, especially considering the global burden of AMR and the limited therapeutic options for DTR pathogens. However, despite the relevance, the manuscript presents methodological weaknesses, statistical inconsistencies, structural problems, and issues with clarity and scientific rigor. Several aspects of the results are difficult to interpret due to convergence problems, model instability, and insufficient methodological description.

Revise for grammar, clarity, and conciseness and remove redundancies.

The cohort includes only 51 patients, yet several multivariable logistic models were constructed.
Given the extremely low events-per-variable , the models are severely underpowered, prone to overfitting, inflated ORs, infinite confidence intervals.

Although Firth correction was used, the manuscript does not discuss its necessity or limitations.

The authors state that targeted therapy with C/A or C/T “improves outcomes,” yet multivariable p-values range from 0.052 to 0.070, confidence intervals are extremely wide, some models failed to converge. These results are suggestive but not statistically robust and should be interpreted with caution.

Given the focus on DTR-PA, it is surprising that: no MIC distributions are provided, no molecular mechanisms are mentioned, no association between resistance phenotype and outcomes is explored. This omission weakens the microbiological relevance of the study.

Pseudomonas aeruginosa must always be italicized.

Comments on the Quality of English Language

The manuscript needs substantial language editing to improve clarity and scientific tone.
Examples: “non-susceptible to all first-line agents strains”. Repetitive expressions (“clinical cure was achieved…”) across sections.

A professional language revision is recommended.

Author Response

Comment 1: "This manuscript investigates predictors of mortality and treatment outcomes in patients with difficult-to-treat resistant Pseudomonas aeruginosa infections. The topic is clinically important, especially considering the global burden of AMR and the limited therapeutic options for DTR pathogens. However, despite the relevance, the manuscript presents methodological weaknesses, statistical inconsistencies, structural problems, and issues with clarity and scientific rigor. Several aspects of the results are difficult to interpret due to convergence problems, model instability, and insufficient methodological description. Revise for grammar, clarity, and conciseness and remove redundancies”.

Response: We acknowledge the reviewer’s concerns, that have been addressed (see below). At any rate, the manuscript has been revised for redundancies, grammar, clarity, and conciseness.

Comment 2: "The cohort includes only 51 patients, yet several multivariable logistic models were constructed. Given the extremely low events-per-variable, the models are severely underpowered, prone to overfitting, inflated ORs, infinite confidence intervals."

Response: We acknowledge the reviewer’s concern regarding the sample size (n=51) relative to the model complexity. DTR-PA is a rare and specific clinical entity, making large single-center cohorts difficult to accrue1. To address the issue of small sample size and "separation" (which causes infinite confidence intervals and inflated ORs in standard logistic regression), we specifically utilized Firth’s bias-reduced penalised likelihood regression Firth’s method is the gold-standard statistical solution for rare event modelling and small sample sizes, as it prevents infinite estimates and reduces small-sample bias. We have revised the Methods section to explicitly justify this choice and clarify that this approach was selected specifically to mitigate the risks of overfitting and instability mentioned by the reviewer. We also stressed the concept in the limitations, but we want to underline that large datasets on DTR-PA in the literature are lacking.

Comment 3: "Although Firth correction was used, the manuscript does not discuss its necessity or limitations."

Response: We agree that further clarification was needed. We have expanded the "Statistical Analysis" section to explain that Firth’s correction was applied specifically to handle the "separation" observed in the data (where certain predictors perfectly predicted the outcome in the standard model) and to provide finite, realistic confidence intervals.

Comment 4: "The authors state that targeted therapy with C/A or C/T “improves outcomes,” yet multivariable p-values range from 0.052 to 0.070, confidence intervals are extremely wide... These results are suggestive but not statistically robust."

Response: This is a fair point. We have tempered the language underlining the borderline significance.

Comment 5: "Given the focus on DTR-PA, it is surprising that: no MIC distributions are provided, no molecular mechanisms are mentioned... This omission weakens the microbiological relevance."

Response: We completely agree that molecular data would strengthen the study. However, due to the retrospective nature of the study (2018–2023) and the routine clinical workflow at our center during that period, molecular resistance mechanisms and cefiderocol testing were not routinely performed. We have added this to the Limitations section, explicitly stating that the lack of molecular characterization and MIC distributions is a limitation of the retrospective design. Please consider that part of the data refers to the COVID-19 period when services were partially disrupted and in the meanwhile C/T was unavailable worldwide for more than a year.

Comment 6: "Revise for grammar, clarity, and conciseness... Pseudomonas aeruginosa must always be italicized."

Response: We have conducted a thorough proofreading of the manuscript to improve English usage and clarity. We have ensured that Pseudomonas aeruginosa and other bacterial names are italicized throughout the text and references.

Reviewer 2 Report

Comments and Suggestions for Authors

The following comments are provided to assist the Editor in making a decision:

  1. The names of bacterial species should be written in italics throughout the manuscript.

  2. The Introduction should be expanded to include more detailed information on the clinical significance of Pseudomonas aeruginosa. Where available, statistical data on infection rates and mortality associated with P. aeruginosa should be reported and appropriately referenced.

  3. The authors should clearly state the guidelines or standards followed for conducting the experiments and data collection, and relevant references should be provided.

  4. For clarity and improved data visualization, the authors are encouraged to include pie chart representations for selected datasets currently presented in tabular form, where appropriate.

  5. Additional information regarding the patient cohort and their characteristics (e.g., demographics, inclusion criteria) should be provided to enhance the transparency and reproducibility of the study.

Author Response

Comment 1: "The names of bacterial species should be written in italics throughout the manuscript."

Response: Thank you. We have corrected the formatting to ensure all instances of Pseudomonas aeruginosa and other species are italicized.

Comment 2: "The Introduction should be expanded to include more detailed information on the clinical significance of Pseudomonas aeruginosa. Statistical data on infection rates and mortality... should be reported."

Response: We have expanded the Introduction to include recent epidemiological data.

Comment 3: "Clearly state the guidelines or standards followed for conducting the experiments and data collection."

Response: We have added a statement in the Methods section confirming that this study was reported in accordance with the STROBE (Strengthening the Reporting of Observational Studies in Epidemiology) guidelines for cohort studies. We also reiterated that the study was approved by the Ethics Committee of the Federico II University Hospital.

Comment 4: "For clarity and improved data visualization, the authors are encouraged to include pie chart representations for selected datasets... Additional information regarding the patient cohort... should be provided."

Response: We appreciate the suggestion to improve visualization. We add Fig. 2 to synthetize the impact of the main variables.

Reviewer 3 Report

Comments and Suggestions for Authors

Clinical Outcomes and Predictors of Mortality in Patients with Difficult-to-Treat Resistant Pseudomonas aeruginosa Infections: A Retrospective Cohort Study. There are multiple limitations in the methodological section of the study, as the authors already mentioned. But it can be improved. These limitations have led to weak statistical analysis.

  1. The study has been performed in a single centre, is retrospective, and may be biased, for the validation of findings with optimised strategies.
  2. The sample size of 51 patients is very small. Also, patients of mixed
  3. new β-Lactam/β-Lactamilase inhibitors (C/T and C/A) were not tested at the beginning of the study. So treatment bias can occur.
  4. Cefiderocol test was not performed.
  5. There should be a broad list of future studies that should include resistance mechanisms.
  6. Invasive device biofilm assessment tests. There should be some standardized procedures for biofilm assessments.
  7. Monotherapy and combination therapy are conflicting. There should be larger, prospective studies to confirm that optimized monotherapy is adequate for critically ill patients.
  8. There are very small number of subjects. There should be a larger sample size for the novel agent to specifically determine whether C/T offers an advantage over C/a.
  9. english need improvement. name should be itlaics.
Comments on the Quality of English Language

Author Response

Comment 1: "The sample size of 51 patients is very small... single centre, retrospective... may be biased."
Response: We fully acknowledge these limitations. We have expanded the Discussion to frankly address the small sample size and single-center design. However, we argue that DTR-PA is a rare phenotype, and even small cohorts provide valuable "real-world" data in an area where randomized controlled trials are difficult to conduct.

Comment 2: "New β-Lactam/β-Lactamase inhibitors (C/T and C/A) were not tested at the beginning of the study. So treatment bias can occur."
Response: This is a correct observation. We have stressed it in the limitations. Please consider that part of the data refers to the COVID-19 period when services were partially disrupted and in the meanwhile C/T was unavailable worldwide for more than a year.

Comment 3: "Cefiderocol test was not performed... There should be a broad list of future studies that should include resistance mechanisms."
Response: We have clarified in the Methods that Cefiderocol testing was not available at our institution during the study period. In the Conclusion, we have added a call for future prospective multicenter studies that integrate molecular resistance mechanisms and biofilm assessment.

Comment 4: "Monotherapy and combination therapy are conflicting... specific determine whether C/T offers an advantage over C/A."

Response: We have expanded the Discussion regarding these controversies.

Comment 5: "Invasive device biofilm assessment tests. There should be some standardized procedures for biofilm assessments."

Response: We agree that biofilm plays a major role, particularly given our finding that device burden predicted failure. We have noted in the limitations that we relied on clinical device burden rather than microbiological biofilm assays.

Comment 6: "There are very small number of subjects"

Response: Please see the answer to comment 1.

Round 2

Reviewer 1 Report

Comments and Suggestions for Authors

Now the manuscript can be published.

Reviewer 2 Report

Comments and Suggestions for Authors

Accept

Reviewer 3 Report

Comments and Suggestions for Authors

Authors have already completed the suggested reviews. Before publication I would suggest a thorough English language and grammar checking of the manuscript. There are multiple places where Pseudomonas aeruginosa is not in italics. and a few other grammatical mistakes. Also, I would suggest changing the format of the tables and making them more presentable and understandable to readers.

Comments on the Quality of English Language